# Characterization of *Pseudomonas kurunegalensis* by Whole-Genome Sequencing from a Clinical Sample: New Challenges in Identification

**DOI:** 10.3390/reports8030104

**Published:** 2025-07-03

**Authors:** David Badenas-Alzugaray, Laura Valour, Alexander Tristancho-Baró, Rossi Núñez-Medina, Ana María Milagro-Beamonte, Carmen Torres-Manrique, Beatriz Gilaberte-Angós, Ana Isabel López-Calleja, Antonio Rezusta-López

**Affiliations:** 1Research Group on Infections Difficult to Diagnose and Treat, Institute for Health Research Aragón, Miguel Servet University Hospital, 50009 Zaragoza, Spain; lvalour@salud.aragon.es (L.V.); aitristancho@salud.aragon.es (A.T.-B.); bgilaberte@salud.aragon.es (B.G.-A.);; 2Clinical Microbiology Service, Miguel Servet University Hospital, 50009 Zaragoza, Spain; 3Area of Biochemistry and Molecular Biology, One Health-UR Research Group, University of La Rioja, 26006 Logroño, Spain

**Keywords:** *Pseudomonas kurunegalensis*, VIM-2, whole-genome sequencing

## Abstract

**Backgoround:** The genus *Pseudomonas* encompasses metabolically versatile bacteria widely distributed in diverse environments, including clinical settings. Among these, *Pseudomonas kurunegalensis* is a recently described environmental species with limited clinical characterization. **Objective and Methods:** In this study, we report the genomic and phenotypic characterization of a *P. kurunegalensis* isolate, Pam1317368, recovered from a catheterized urine sample of a post-renal transplant patient without symptoms of urinary tract infection. Initial identification by MALDI-TOF MS misclassified the isolate as *Pseudomonas monteilii*. Whole-genome sequencing and average nucleotide identity (ANI) analysis (≥95%) confirmed its identity as *P. kurunegalensis*. The methodology included genomic DNA extraction, Illumina sequencing, genome assembly, ANI calculation, antimicrobial susceptibility testing, resistance gene identification and phylogenetic analysis. **Results:** Antimicrobial susceptibility testing revealed multidrug resistance, including carbapenem resistance mediated by the metallo-β-lactamase gene *VIM-2*. Additional resistance determinants included genes conferring resistance to fluoroquinolones and aminoglycosides. Phylogenetic analysis placed the isolate within the *P. kurunegalensis* clade, closely related to environmental strains. **Conclusions:** Although the clinical significance of this finding remains unclear, the presence of clinically relevant resistance genes in an environmental *Pseudomonas* species isolated from a human sample highlights the value of genomic surveillance and accurate species-level identification in clinical microbiology.

## 1. Introduction

The genus *Pseudomonas* comprises Gram-negative, non-glucose-fermenting, rod- shaped bacteria belonging to the order *Pseudomonadales*, family *Pseudomonadaceae*, and includes a broad diversity of species widely distributed in natural and anthropogenically impacted environments. Members of this genus are recognized for their remarkable metabolic versatility and adaptability, which allows them to thrive in soil, water, plant surfaces, and even hospital settings such as drains [1], cleaning products, and medical equipment. Notably, *Pseudomonas* aeruginosa has emerged as a major opportunistic human pathogen, frequently implicated in healthcare-related infections such as ventilator-associated pneumonia, urinary tract infections, and bacteraemia, particularly in immunocompromised individuals [2]. These infections are often difficult to treat due to intrinsic and acquired resistance to multiple antibiotic classes, including β-lactams, aminoglycosides, and fluoroquinolones [3], underscoring the clinical importance of this genus.

The ecological niche of *Pseudomonas* species is notably broad, ranging from terrestrial and aquatic ecosystems to associations with plants, animals, and humans [4]. Many species are known for their plant-growth-promoting properties or biocontrol activities, while others contribute to bioremediation through the degradation of pollutants [5]. This ubiquity increases the opportunities for horizontal gene transfer, including antibiotic resistance determinants, which can spread between environmental and clinical strains [6].

*Pseudomonas kurunegalensis* is a relatively recently described species, initially isolated from soil in Sri Lanka in 2022. It belongs to the well-characterized *Pseudomonas putida* group, a cluster of environmental bacteria frequently encountered in clinical microbiology laboratories. Due to the limitations of routine identification methods such as MALDI-TOF MS, isolates are often reported only at the group level. Although less studied than *Pseudomonas aeruginosa*, *P. kurunegalensis* shares several traits typical of the *P. putida* group, including the ability to degrade aromatic compounds and adapt to a wide range of ecological niches, highlighting its environmental resilience [7]. Preliminary genomic studies have identified putative resistance genes and efflux systems in its genome, suggesting an intrinsic capacity to withstand various antimicrobial agents [8]. The presence of such genes in an environmental context raises concerns regarding the potential mobilization of resistance factors from environmental *Pseudomonas* species to clinically relevant pathogens, underscoring the importance of detailed genomic characterization of clinical isolates [6].

Accurate identification of lesser-known *Pseudomonas* species remains challenging due to high genetic similarity within the genus. Conventional molecular tools, including 16S rRNA gene sequencing, often lack sufficient resolution to distinguish closely related species. In recent years, more robust approaches—such as whole-genome sequencing (WGS) and average nucleotide identity (ANI)—have become essential for bacterial taxonomy. WGS provides comprehensive information on the organism’s genetic content, while ANI quantifies the overall sequence similarity between genomes, enabling species-level classification with high precision. These tools are particularly valuable when initial identification by standard methods is inconclusive or misleading [9].

Despite the increasing recognition of non-*aeruginosa Pseudomonas* species in clinical settings, the role and clinical impact of *P. kurunegalensis* remain poorly understood. To date, there is one report of this species being isolated from human clinical samples. This represents a gap in our understanding of the epidemiology and resistance dynamics of emerging environmental *Pseudomonas* species that may enter clinical environments.

This study aims to characterize a *P. kurunegalensis* isolate, Pam1317368, obtained from a clinical sample, initially misidentified as *Pseudomonas monteilli*, at the genomic level and explore its antimicrobial resistance profiles.

## 2. Material and Methods

### 2.1. Sample Collection and Microbiological Processing

The microorganism was isolated from a urine sample collected during a post-renal transplantation monitoring protocol. The sample was processed according to the standardized protocols of the Microbiology Laboratory at Miguel Servet University Hospital. This urine protocol included semi-quantitative inoculation of 10 μL of a non-selective chromogenic medium ORI CHROMAGAR ORIENTATION (Becton Dickinson^®^, Franklin Lakes, NJ, USA) and 24 h incubation at 35 °C in an aerobic atmosphere.

Microbial identification was carried out using matrix-assisted laser desorption/ionization time-of-flight mass spectrometry (MALDI-TOF MS) (MALDI Biotyper™, Bruker^®^, Bremen, Germany) in accordance with the manufacturer’s instructions [10].

Antimicrobial susceptibility testing was conducted using a MicroScan™ NEG MIC 57 panel (Catalog No. C58012; Beckman Coulter^®^, Brea, CA, USA) and the EUCAST disk diffusion method (employing MH agar Biomérieux^®^, Marcy-l’Étoile, France, and OXOID Termofisher^®^ disks, Waltham, MA, USA). The results were interpreted according to the EUCAST criteria v15.0 for *Pseudomonas* spp., except for Ceftolozane/tazobactam, Ceftazidime/avibactam, and cefiderocol, where *P. aeruginosa* criteria were applied, given the lack of specific breakpoints for other *Pseudomonas* species [11].

### 2.2. Genomic Sequencing Analysis

Prior to genomic DNA extraction, the isolate was subcultured on blood agar and incubated under aerobic conditions for 24 h to ensure culture purity and optimize biomass yield. Genomic DNA was extracted from a pure bacterial culture using a bead-based nucleic acid extraction protocol on a MagCore^®^ system (RBC Bioscience^®^, New Taipei City, Taiwan). DNA concentration was quantified using a Qubit™ fluorometer (Thermo Fisher Scientific^®^, Walthman, MA, USA), and libraries were prepared with a Nextera XT DNA Library Preparation Kit (Illumina^®^, San Diego, CA, USA). Whole-genome sequencing was performed on the Illumina MiSeq™ platform employing a 150 bp paired-end strategy (300 cycles). Raw sequencing data quality was assessed with FastQC [12] v0.12.1 and subjected to trimming with Trimmomatic [13] v0.39. De novo genome assembly was conducted using Unicycler [14] v0.5.0, and potential contamination was evaluated with GUNC [15] v1.0.5. Assembly quality was assessed with QUAST [16] v5.2.0. Annotation of the draft genome was performed using Prokka [17] v1.14.6.

Species identification was conducted through a multi-step approach combining gene-based and genome-wide methods. The sequences of 16S rRNA, *rpoD* (RNA polymerase sigma factor RpoD), and *gyrB* (DNA gyrase subunit B) were extracted from the annotated genome and compared against the BLAST v2.16.0 core_nt database. Complementary genome-level identification was performed using GAMBIT [18] v1, Mash [19] v2.1, and the PubMLST [20] and fIDBAC [21] species identification services, as well as the Type Strain Genome Server (TYGS) [22].

An ANI-based dendrogram was constructed using *dRep* [23] v3.4.5, comparing the isolate Pam1317368 against a subset of *Pseudomonas* genomes identified as the most similar by preliminary identification tools. A threshold of ≥95% ANI was applied for species delineation [24].

Phylogenetic analysis based on whole-genome SNPs (single-nucleotide polymorphisms) was carried out using Snippy [25] v4.6.0 and *IQ-TREE* [26] v2.3.6, with support estimated through 1000 ultrafast bootstrap replicates. The genome of *P. kurunegalensis* NY4817 was used as a reference and *Azotobacter vinelandii* DJATCC BAA1303 as an outgroup. Recombination events were identified and filtered using Gubbins [27] v3.4. The resulting maximum likelihood tree was visualized using the Interactive Tree Of Life iTOL [28] v7.0.

In parallel, a heatmap combining ANI values and phylogenetic relationships between Pam1317368 and 15 closely related *Pseudomonas* reference genomes was generated with ANIclustermap [29] v1.4.0.

### 2.3. Resistome Characterization

Antimicrobial resistance gene profiling was annotated using the Resistance Gene Identifier RGI [30] v6.0.3 in conjunction with the Comprehensive Antibiotic Resistance Database CARD v3.2.9, as well as ABRicate [31] v1.0.1, employing its card resistance gene database v3.2.9.

Plasmid reconstruction was performed using MOB-Recon [32] v3.1.9.

## 3. Results

### 3.1. Clinical Case Description

A 79-year-old male patient underwent a renal transplant. After the surgery, a double-J stent and a bladder catheter were placed. The renal post-transplant protocol included serial urine cultures at days +1, +2, +3, +5, and +8. The first four cultures obtained during the initial days of observation showed no microbial growth. However, the fifth urine culture revealed the presence of Gram-negative bacilli at 60,000 colony-forming units (CFU)/mL. At the time of sampling, the patient was asymptomatic and was not hospitalized.

Initial identification of the isolate by MALDI-TOF MS yielded *P. monteilii* with a score of 2.07.

### 3.2. Antimicrobial Susceptibility Testing and Carbapenemase Detection

According to EUCAST interpretative criteria, the following results were obtained (Table 1).

Given the observed resistance to carbapenems, an NG-Test^®^ CARBA-5 assay (NG Biotech^®^ Guipry-Messac, France), designed to detect the five major carbapenemase families (*NDM*, *IMP*, *VIM*, *OXA-48*, and *KPC*), was performed to assess the potential presence of plasmid-mediated carbapenemases, yielding a positive result for *VIM*.

In accordance with hospital protocols following the detection of a *VIM*-type carbapenemase, the isolate underwent whole-genome sequencing.

### 3.3. WGS Analysis

The sequencing process yielded an average coverage of 62X. Assembly quality metrics indicated a total of 69 contigs, with the longest contig measuring 781,549 base pairs (bp). The assembly presented an N50 value of 199,128 bp and an L50 of nine contigs. The draft genome had an estimated size of 5.8 Mb and included 5378 coding sequences, 4 rRNA genes, 1 tmRNA, and 46 tRNA.

Identification based on BLAST analysis of the 16S rRNA gene revealed 100% sequence identity with *P. putida*, *P. monteilli*, and *P. kurunegalensis*, preventing species-level discrimination. Further analysis of housekeeping genes supported identification as *P. putida group*, with both *rpoD* and *gyrB* genes showing 100% identity to *P. monteilii* and *P. kurunegalensis*, thereby highlighting the close genetic similarity between these two species and complicating species-level discrimination.

While genome-wide similarity tools such as GAMBIT and Mash suggested correspondence with *P. monteilii* and *Pseudomonas* spp., respectively, alternative tools including PubMLST, fIDBAC, and the Type Strain Genome Server (TYGS) consistently identified the isolate as *P. kurunegalensis*.

To resolve the discrepancies, an analysis of genomic relatedness was performed. An ANI-based dendrogram placed the isolate within the *P. kurunegalensis* clade, showing 99.96% identity to the reference genome *P. kurunegalensis* NY4817. Phylogenetic analysis based on SNPs confirmed this classification, as the isolate formed a monophyletic clade with other *P. kurunegalensis* strains (Figure 1). To further resolve species-level classification, a dendrogram based on ANI supported the assignment, clustering the isolate within the *P. kurunegalensis* clade (Figure 2). Taken together, these results provide robust evidence to confirm the taxonomic identification of Pam1317368 as *P. kurunegalensis.*

The resistome annotation of isolate Pam1317368 revealed a range of resistance mechanisms present in its genome. However, the analysis was limited to mechanisms directly involved in the enzymatic modification of antibiotic molecules or alterations at their target sites. Resistance mechanisms related to changes in membrane permeability, active efflux pumps, or modifications in the lipopolysaccharide profile were intentionally excluded from the scope of the analysis due to low percentages of identity.

Among the mechanisms considered, resistance to fluoroquinolones was associated with the presence of the *QnrVC6* gene and a variant of the *gyrA* gene. The *QnrVC6* gene encodes a pentapeptide repeat protein that protects DNA gyrase and topoisomerase IV from inhibition by fluoroquinolones, thereby preserving bacterial DNA replication in the presence of the antibiotic [33]. The resistance-associated mutation identified in the *gyrA* gene, T83I, confers resistance to fluoroquinolones as well. This mutation involves a substitution of threonine (T) with isoleucine (I) at amino acid position 83 of the *gyrA* protein, a subunit of DNA gyrase. This enzyme is the primary target of fluoroquinolones, and the T83I mutation alters the quinolone binding site, reducing drug affinity and thereby diminishing the antibiotic’s effectiveness. The sequence identity of the *gyrA* gene was 85.27%, which is relatively low due to the reference sequence in the database corresponding to *Pseudomonas aeruginosa*, whereas Pam1317368 belongs to the *Pseudomonas putida* group [34].

For aminoglycosides, the genes APH(6)-Id [35], AAC(6′)-Ib9, and AAC(6′)-IIa [36] were identified. The APH(6)-Id gene encodes an aminoglycoside phosphotransferase that inactivates the antibiotic by phosphorylation. AAC(6′)-Ib9 and AAC(6′)-IIa encode aminoglycoside acetyltransferases, which modify the drugs by acetylation, thereby preventing their binding to the bacterial ribosome and impairing their bactericidal activity. These genes showed sequence identities ranging from 99.42% to 100%.

In addition, the β-lactamase gene *VIM-2* was detected with 100% identity. This gene encodes a metallo-β-lactamase capable of hydrolyzing a broad range of β-lactam antibiotics, including carbapenems, rendering them ineffective [37].

Finally, the *qacEΔ1* gene, identified with complete identity, is associated with resistance to disinfectants and antiseptics, particularly quaternary ammonium compounds. It encodes an efflux pump component that contributes to reduced susceptibility to biocidal agents commonly used in hospital environments [38].

No plasmid was detected; therefore *VIM-2* is chromosomal.

### 3.4. Clinical Outcome

The patient received empirical treatment with ciprofloxacin until antimicrobial susceptibility testing results became available. Upon confirmation of resistance to the drug, ciprofloxacin was discontinued. In the context of asymptomatic bacteriuria caused by an extensively drug-resistant (XDR) microorganism, a consensus was reached between the Antimicrobial Stewardship Program (PROA) team and the Urology Department to remove the urinary catheter and subsequently obtain a follow-up urine culture. Successive microbiological controls were negative, and the double-J ureteral stent was removed ten days later.

## 4. Discussion

The genus *Pseudomonas* is renowned for its environmental adaptability and remarkable genomic plasticity, enabling many of its members to colonize diverse ecological niches, including clinical environments [1]. Within this genus, the *P. putida* group has received increasing attention due to its metabolic diversity and biotechnological potential, but also for its emerging clinical relevance. Traditionally considered non-pathogenic, species within this group have recently been reported in hospital settings, occasionally associated with multidrug resistance (MDR) phenotypes and mobile resistance elements [9,39].

Urinary tract infections (UTIs) are among the most frequent infectious complications in kidney transplant recipients, particularly during the early post-transplant period. Reported incidence rates range from 42% to 75%, depending on factors such as applied definitions, diagnostic criteria, study design, and length of follow-up [40]. While *Escherichia coli* remains the most prevalent uropathogen, the genus *Pseudomonas*, though less commonly isolated, holds significant clinical relevance due to its intrinsic resistance to multiple classes of antibiotics.

In this context, our study reports a clinical isolation of *P. kurunegalensis*, a species first delineated in Sri Lanka as part of a large-scale phylogenomic study redefining the *P. putida* group. The isolate was obtained from the urine of a 79-year-old patient. While this finding highlights the occurrence of environmental *Pseudomonas* species in clinical samples, it is important to consider that isolation from a single urine culture, especially from a catheterized specimen, might represent contamination rather than true colonization of a clinical site. Potential sources of contamination include water condensation in the urine collection cup, environmental contamination within the microbiology laboratory, or disinfectant residues during sample processing. After the J stent was replaced, the removed device was not reanalyzed; however, the patient tested negative following the replacement. Regardless, the detection of this isolate carrying clinically relevant resistance determinants underscores the need for careful interpretation and continued genomic surveillance of environmental *Pseudomonas* in clinical settings.

Comparative genomic analyses based on ANI confirmed that Pam1317368 should be assigned and reported as *P. kurunegalensis*, showing 99.96% identity with *P. kurunegalensis* NY4817. The next closest strain was *P. kurunegalensis* ANKCG2, with 96.39% identity. ANI values around 95% were observed with *P. monteilii*, while all other Pseudomonas species showed identities below this threshold, and certain strains classified as *P. kurunegalensis* exhibit lower genomic similarity to Pam1317368 than *Pseudomonas monteilii.* This finding may reflect underlying heterogeneity within the *P. kurunegalensis* group, or alternatively, ambiguous species assignments in current reference databases. This finding underscores the limitations of conventional identification methods such as MALDI-TOF and 16S rRNA sequencing, which have been shown to frequently misclassify species within the *P. putida* group [41]. Notably, Pam1317368 was initially misidentified as *P. monteilii*, a member of the *P. putida* group, by MALDI-TOF, emphasizing the importance of genome-based approaches for accurate bacterial identification, especially in critically ill or vulnerable patients.

Following a comprehensive analysis of the available genomic and phylogenetic data, the isolate was identified as *P. kurunegalensis*. This species, although rarely reported in clinical settings, was found to harbor the *VIM-2* gene with 100% identity, a metallo-β-lactamase known to confer resistance to carbapenems and commonly associated with plasmid-mediated dissemination among Gram-negative pathogens. The detection of *bla_VIM-2_* in this isolate underscores the growing threat posed by multidrug-resistant *Pseudomonas* spp. in healthcare settings, particularly in immunocompromised patients, such as our kidney transplant recipient.

Although *VIM-2* is frequently associated with plasmid-borne class 1 integrons, recent evidence confirms its stable chromosomal integration in members of the *P. putida* group. In particular, a recent study demonstrated the presence of a chromosomally encoded class 1 integron carrying *VIM-2* in *Pseudomonas alloputida*, indicating that this resistance determinant may become stably integrated into the chromosome of environmental or clinical isolates of *Pseudomonas* [42]. This suggests that the dissemination of *VIM-2* in the *P. putida* group may not depend exclusively on plasmid mobility but also on chromosomal reservoirs.

Previous reports have documented the emergence of *VIM-2*-producing *Pseudomonas putida* strains in regions such as Brazil and Argentina, frequently in the context of hospital outbreaks and among patients with limited therapeutic alternatives [40,43]. In these cases, treatment regimens have included colistin, amikacin, ciprofloxacin, and gentamicin. For instance, combinations such as meropenem plus amikacin or meropenem plus colistin have been used to manage infections caused by carbapenem-resistant *Pseudomonas* spp., although success rates have depended on infection site, bacterial load, and the host’s clinical status [44,45]. It is important to note that the isolate was recovered from colonization rather than active infection. Nevertheless, accurate species identification and characterization remain relevant due to their epidemiological significance. Colonizing strains harboring resistance genes such as bla_VIM-2_ may serve as reservoirs within healthcare environments and represent a potential risk for subsequent infection, especially in immunocompromised patients or those with deteriorating clinical conditions. Early detection and surveillance are therefore essential to guide infection control measures and inform appropriate therapeutic decisions if infection develops.

The patient’s antimicrobial treatment initially included ciprofloxacin. This strategy proved effective, and the patient did not experience any additional symptoms during follow-up. After reviewing the antimicrobial resistance results, a consultation with the Infectious Diseases team was conducted, and it was decided to continue with a conservative management approach, which included the removal of the urinary catheter. This case underscores the importance of conducting detailed antimicrobial resistance analyses to guide therapeutic decisions and highlights the need for continuous surveillance in patients with urinary tract infections, particularly those with invasive devices such as catheters.

Given the limited number of active agents against *VIM*-producing strains, therapeutic selection must be guided by in vitro susceptibility testing. Importantly, the nephrotoxicity associated with colistin and aminoglycosides warrants caution, especially in vulnerable populations such as kidney transplant recipients, where renal function is already compromised and therapeutic options are scarce.

The use of WGS and ANI is limited in routine diagnostics due to high costs, long turnaround times, and limited accessibility in many clinical laboratories. This complicates accurate identification in real-time clinical decision-making and can lead to misclassification or underreporting of rare or emerging *Pseudomonas* species. Moreover, current reference databases often lack comprehensive genomic representation of lesser-known environmental species, leading to ambiguous or erroneous species assignments. This has been highlighted in recent reports of environmental *Pseudomonas* species isolated from clinical settings, where 16S rRNA and MALDI-TOF results may align but still be insufficient for confident classification in the absence of high-quality reference genomes [46]. Additionally, the relatively low sequencing quality of the present study, reflected by the assembly consisting of 69 contigs and a low N50, may limit the resolution and reliability of genomic analyses. These challenges underscore the importance of integrating genomic data into taxonomic efforts and clinical microbiology, particularly when environmental bacteria are implicated in infections.

## 5. Conclusions

Accurate identification of *Pseudomonas kurunegalensis* in clinical settings represents a significant challenge due to its high genetic similarity with closely related species, which can lead to misidentification by conventional methods such as MALDI-TOF MS. This study demonstrates that whole-genome sequencing analysis, including average nucleotide identity (ANI) comparisons and SNP-based phylogenetic reconstruction, is essential for correct species-level classification. The detection of a multidrug-resistant *P. kurunegalensis* isolate carrying the *VIM-2* carbapenemase gene illustrates the potential for environmental species within the *P. putida* group to harbor clinically relevant resistance determinants. However, given the absence of effective antimicrobial treatment in this case and the favorable clinical outcome following removal of the urinary catheter and microbiological monitoring, the pathogenic role of this isolate remains uncertain. This underscores the complexity of interpreting such findings, especially in patients with indwelling devices where contamination or colonization cannot be ruled out. The broad antimicrobial resistance profile observed, including genes conferring resistance to β-lactams, fluoroquinolones, and aminoglycosides, underscores the need for continuous and stringent surveillance in hospital environments to prevent the spread of such strains. Additionally, the presence of the *qacEΔ1* gene suggests potential resistance to disinfectants commonly used in healthcare settings, which could complicate infection control strategies. Finally, the clinical outcome of this case showed that removal of the urinary catheter combined with appropriate microbiological monitoring was effective for infection resolution, emphasizing the importance of comprehensive clinical management in the face of multidrug-resistant microorganisms.

## Figures and Tables

**Figure 1 reports-08-00104-f001:**
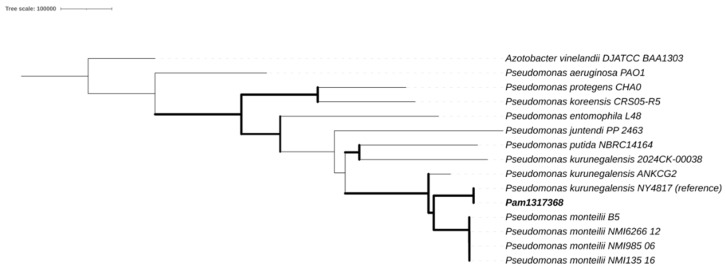
Core single-nucleotide-polymorphism-based maximum likelihood phylogeny.

**Figure 2 reports-08-00104-f002:**
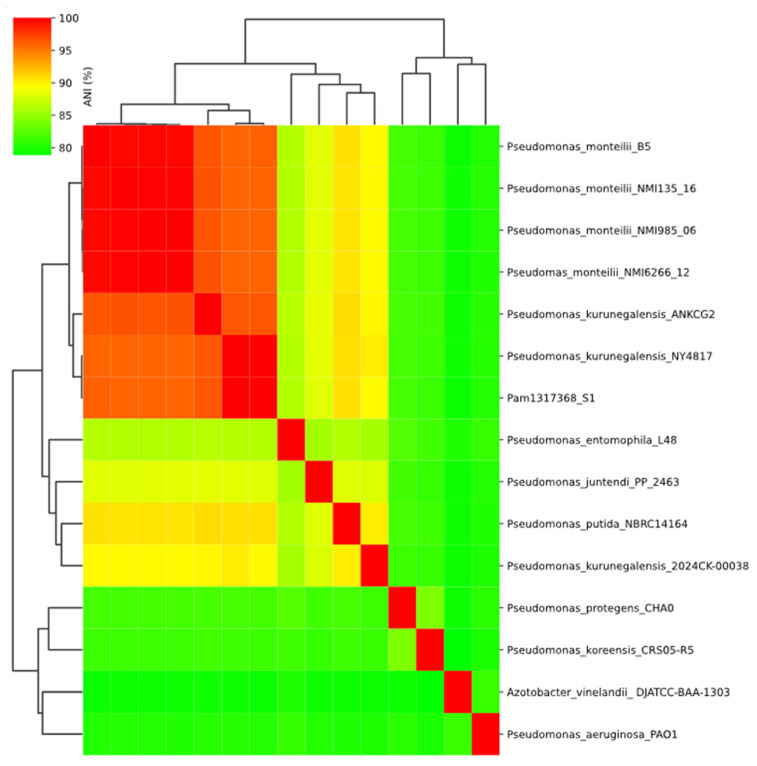
ANI heatmap.

**Table 1 reports-08-00104-t001:** Antimicrobial susceptibility according to EUCAST criteria.

	Disk Diffusion Method		Microscan Panel		
Antibiotic (Disc Content Mcg)	Zone Diameter (mm)	Zone Diameter Breakpoint (mm) R<	MIC (mg/L)	MIC Breakpoint (mg/L) R>	Interpretation
Ceftazidime (10)	12	17	32	8	R
Cefepime (30)	17	21	>8	8	R
Ceftazidime/avibactam (*P. aeruginosa* criteria) (10-4)	12	17	>4	8	R
Ceftolozane/tazobactam (*P. aeruginosa* criteria) (30-10)	6	23	>8	4	R
Cefiderocol (*P. aeruginosa* criteria) (30)	28	22	≤0.016	2	S
Piperacillin-tazobactam (30-6)	12	18	>16	16	R
Aztreonam (30)	21	18	16	16	I
Imipenem (10)	6	20	>4	4	R
Meropenem (10)	6	18	>8	2	R
Ciprofloxacin (5)	6	26	>1	0.5	R
Levofloxacin (5)	6	18	>1	2	R
Tobramycin (10)	13	18	>4	2	R
Amikacin (30)	26	15	≤8	16	S
Colistin			≤2	4	S

## Data Availability

The genome was submitted to GenBank on 16 June 2025 under BioSample ID SAMN48908813.

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
