# Peer review of "Characterization of Pseudomonas kurunegalensis by Whole-Genome Sequencing from a Clinical Sample: New Challenges in Identification"

_reports, 2025, doi:10.3390/reports8030104_

Round 1
Reviewer 1 Report
Comments and Suggestions for Authors
The authors of the mansucript entitled - Characterization of Pseudomonas kurunegalensis by Whole-Genome Sequencing from a Clinical Sample: New Challenges in Identification – describe in their manuscript the characterization of an environmental bacterium with well founded methodology. The bacterium was just recently recognized as an independent species within the ever expanding Genus Pseudomonas, based on genomic delineation. Therfore the report adds to the literature a description of that specific bacterium in a clinical context with a focus on identification problems which is an interesting topic. This delineation is quite new and the number of reference genomes with available metadata is therefore limited. Also, the description of acquired resistance in environmental bacteria by mobile genetic elements is a timely and relevant topic. Overall, the manuscript merits publication, but I have some comments for potential improvement.
Major comments:
- Talk down
At most points the authors are descriptive and correct by sticking to facts. On the other hand I found some lines in the draft that are questionable and may conclude too much from the described occasion, that the isolate was found in one urine sample from one patient at one time:
Abstract:
- I do not agree, that kurunegalensis is an „emerging opportunistic pathogen with extensive resistance traits“. In general one should not state that based on a single observation. If the VIM-2 was not present I would consider it totally reasonable to not care about this environmental pseudomonas at all, even if found in urine, which could also represent contamination. Emerging pathogen is an (unfortunately) too frequently used term and not well defined, but I strongly recommend to not use it for this occasion here.
- „although its clinical relevance remains to be fully elucidated“ ## From my point of view kurunegalensis is not a new bacterium and it is therfore not correct to think, that we don’t know anything about it yet. It is just a semantic thing based on genomic delineation. In many clinical microbiology laboratories it is common to report such pathogens group-related (e.g. MALDI_TOF reports P. monteilii, the microbiologist knowing about the limitations of the identification technique just reports P. putida group to the clinicians). I think we already know a lot about the P. putida group and we also know that members of this group may acquire mobile genetic elements and may be found in urine without any clinical significance. I invite the authors to think about that more sticking in this particular case but also in the whole manuscript
Discussion:
- „not only confirms the potential of environmental Pseudomonas species to colonize clinical niches“ ## Especially if the isolate was isolated from one culture out of many, it could also be part of the Urine sample as contamination rather than colonizing clinical niches (analytical or preanalytical contamination (water condensation in the urine cup, water in the microbiology lab, liquids desinfectants throughout the diagnostic workflow...). If the isolate was present on the J stent one would expect more findings than one. Was the J stent analysed after replacement?
Conclusion:
- „The detection of a multidrug-resistant clinical isolate of P. kurunegalensis carrying the VIM-2 carbapenemase gene highlights the emerging potential of this environmental species as an opportunistic pathogen, particularly in immunocompromised patients or those with indwelling medical devices.“ Again, talk down especially as the related clinical decision contradicts the statement (no effective antimicrobial treatment performed)
Minor comments in order of appearance:
Introduction) Please check the introduction for redundant statements. I feel the introduction is quite long and general for this type of article that describes only a single report of a bacterial species in a somehow novel context (e.g. the word environment/ environmental is used about ten times within the introduction, just think about if it could be enough to mention that one time?). In contrast the authors could add some introductional statements for the reader, with respect to WGS and ANI. It appears to me they chose the ANI results to be robust enough to clear the not too descriminating results from BLAST of the raw data. Despite this being major for the study and the identification problem, it is not introduced in the introduction in contrast to the repeated general information about Pseudomonas. Many readers may not be familiar with these terms as the journal is not particularly designated to geneomics.
Introduction) The authors emphasize a very relevant point here: „While molecular tools such as 16S rRNA gene sequencing provide a foundation, the resolution of species boundaries often requires more robust approaches, including multilocus sequence analysis, average nucleotide identity (ANI), and whole genome sequencing“. One could add the fact that these techniques are costly and mostly unavailable and also too time consuming for clinical decision making. This complicates identification in routine diagnostics which emphasizes the relevance of the study the authors conducted here. It could additionally be emphasized by comparison and reference to other studies/ authors, e.g. reports about other environmental Pseudomonas with limitations in identification (e.g. sequencing discussed as potentially being too imprecise in a relevant infection with an environmental Pseudomonas despite correlation of 16S and MALDI-TOF https://doi.org/10.3390/idr16040053). The problem is in fact very relevant for routine, this could be made clearer to the reader in the introduction or in the discussion by referring to similar descriptions. The limitations of the databases in rare species („ambiguous species assignments in current reference databases.“ Reference? and or indicate how many reference genomes you used?) is a major point that is also discussed by these and other authors for rare species and is currently not referenced in the article here.
Methods) Cefiderocol should not be performed with MIC Test strips, the EUCAST warns to do this (https://www.eucast.org/ast-of-bacteria/warnings), I recommend to repeat susceptibility testing (AST) for cefiderocol and to clarify for the whole manuscript which guidelines were used to perform AST (with respect to all results).
Methods) Provide reference for the decision: „A threshold of ≥ 95% ANI was applied for species delineation“
Results) I find it odd to describe the clinical case behind the isolate isolation without heading as a start of the results section
Results) „However, the fifth urine culture revealed the presence of Gram-negative bacilli at 60,000 colony-forming units (CFU)/ml.“ What about the culture after that? The last paragraph of section 3 does not help as it is unclear whether the cultures here were from the protocol (+8) or part of clinical decisions after the protocol
Results) „Colistin (MIC ≤2 mg/L)“, See above, Colistin is also special with respect to AST, please indicate which guidelines were followed and how Colistin testing was performed
Whole manuscript) Spelling of genes should be consistent within the draft (e.g. different usage of bla and hyphen) and consistent with the international nomenclature (double check for italics mistakes, there are some)
Author Response
|
1. Summary |
|
|
|
Thank you very much for taking the time to review this manuscript. Please find the detailed responses below and the corresponding revisions/corrections highlighted/in track changes in the re-submitted files.
|
||
|
2. Point-by-point response to Comments and Suggestions for Authors |
||
|
Comments 1: Abstract: I do not agree, that kurunegalensis is an „emerging opportunistic pathogen with extensive resistance traits“. In general one should not state that based on a single observation. If the VIM-2 was not present I would consider it totally reasonable to not care about this environmental pseudomonas at all, even if found in urine, which could also represent contamination. Emerging pathogen is an (unfortunately) too frequently used term and not well defined, but I strongly recommend to not use it for this occasion here.
|
||
|
Response 1: Thank you for pointing this out. We agree with this comment. The phrase that pseudomonas kurunegalensis is an emerging pathogen is eliminated. In the new version of the manuscript in the abstract you can see this change.
|
||
|
Comments 2: Introduction although its clinical relevance remains to be fully elucidated“ ## From my point of view kurunegalensis is not a new bacterium and it is therfore not correct to think, that we don’t know anything about it yet. It is just a semantic thing based on genomic delineation. In many clinical microbiology laboratories it is common to report such pathogens group-related (e.g. MALDI_TOF reports P. monteilii, the microbiologist knowing about the limitations of the identification technique just reports P. putida group to the clinicians). I think we already know a lot about the P. putida group and we also know that members of this group may acquire mobile genetic elements and may be found in urine without any clinical significance. I invite the authors to think about that more sticking in this particular case but also in the whole manuscript
|
||
|
Response 2: Thank you for pointing this out. We have, accordingly, modified this part. Eliminating the part that says "although its clinical relevance remains to be fully elucidated" This can be found between lines 55 and 62.
|
||
|
Comments 3: „not only confirms the potential of environmental Pseudomonas species to colonize clinical niches“ ## Especially if the isolate was isolated from one culture out of many, it could also be part of the Urine sample as contamination rather than colonizing clinical niches (analytical or preanalytical contamination (water condensation in the urine cup, water in the microbiology lab, liquids desinfectants throughout the diagnostic workflow...). If the isolate was present on the J stent one would expect more findings than one. Was the J stent analysed after replacement?
|
||
|
Response 3: Thank you for pointing this out. Perfect, these modifications are between lines 261 and 265.
|
||
|
Comments 4: Conclusions „The detection of a multidrug-resistant clinical isolate of P. kurunegalensis carrying the VIM-2 carbapenemase gene highlights the emerging potential of this environmental species as an opportunistic pathogen, particularly in immunocompromised patients or those with indwelling medical devices.“ Again, talk down especially as the related clinical decision contradicts the statement (no effective antimicrobial treatment performed)
|
||
|
Response 4: Thank you for pointing this out ,we agree with this comment, and have therefore modified the conclusions. These modifications are found between lines 349 and 356.
|
||
|
Comments 5: Introduction) Please check the introduction for redundant statements. I feel the introduction is quite long and general for this type of article that describes only a single report of a bacterial species in a somehow novel context (e.g. the word environment/ environmental is used about ten times within the introduction, just think about if it could be enough to mention that one time?). In contrast the authors could add some introductional statements for the reader, with respect to WGS and ANI. It appears to me they chose the ANI results to be robust enough to clear the not too descriminating results from BLAST of the raw data. Despite this being major for the study and the identification problem, it is not introduced in the introduction in contrast to the repeated general information about Pseudomonas. Many readers may not be familiar with these terms as the journal is not particularly designated to geneomics.
|
||
|
Response 5: Thank you for pointing this out .The number of times the terms "environment" or "environmental" appeared in the introduction was reduced. We've also added an explanation of the terms ANI and WGS. These changes are found between lines 69 and 77.
|
||
|
Comments 6: Introduction) The authors emphasize a very relevant point here: „While molecular tools such as 16S rRNA gene sequencing provide a foundation, the resolution of species boundaries often requires more robust approaches, including multilocus sequence analysis, average nucleotide identity (ANI), and whole genome sequencing“. One could add the fact that these techniques are costly and mostly unavailable and also too time consuming for clinical decision making. This complicates identification in routine diagnostics which emphasizes the relevance of the study the authors conducted here. It could additionally be emphasized by comparison and reference to other studies/ authors, e.g. reports about other environmental Pseudomonas with limitations in identification (e.g. sequencing discussed as potentially being too imprecise in a relevant infection with an environmental Pseudomonas despite correlation of 16S and MALDI-TOF https://doi.org/10.3390/idr16040053). The problem is in fact very relevant for routine, this could be made clearer to the reader in the introduction or in the discussion by referring to similar descriptions. The limitations of the databases in rare species („ambiguous species assignments in current reference databases.“ Reference? and or indicate how many reference genomes you used?) is a major point that is also discussed by these and other authors for rare species and is currently not referenced in the article here.
|
||
|
Response 6: Thank you for pointing this out. We thought about adding this part to the discussion, instead of the introduction. This can be found between lines 294 and 301.
|
||
|
Comments 7: Methods) Cefiderocol should not be performed with MIC Test strips, the EUCAST warns to do this (https://www.eucast.org/ast-of-bacteria/warnings), I recommend to repeat susceptibility testing (AST) for cefiderocol and to clarify for the whole manuscript which guidelines were used to perform AST (with respect to all results).
|
||
|
Response 7: Thank you for pointing this out. You are right. These changes can be found between lines 98 and 107.
|
||
|
Comments 8: Methods) Provide reference for the decision: „A threshold of ≥ 95% ANI was applied for species delineation“
|
||
|
Response 8: Perfect, we have added the bibliographic reference where this threshold appears.
Richter, M. & Rosselló-Móra, R. Shifting the genomic gold standard for the prokaryotic species definition. Proc. Natl. Acad. Sci. 106, 19126–19131 (2009).
Found at line 129-130
|
||
|
Comments 9: Results) I find it odd to describe the clinical case behind the isolate isolation without heading as a start of the results section
|
||
|
Response 9: We agree, we have added sections to the results. 3.1 Clinical case description 3.2 Antimicrobial Susceptibility Testing and Carbapenemase Detection 3.3 WGS analysis 3.4 Clinical outcome
|
||
|
Comments 10: Results) „However, the fifth urine culture revealed the presence of Gram-negative bacilli at 60,000 colony-forming units (CFU)/ml.“ What about the culture after that? The last paragraph of section 3 does not help as it is unclear whether the cultures here were from the protocol (+8) or part of clinical decisions after the protocol
|
||
|
Response 10: The patient was treated with Ciprofloxacin and a sample was taken again later and nothing was isolated. You can find this information between lines 231 and 238.
|
||
|
Comments 11: Results) „Colistin (MIC ≤2 mg/L)“, See above, Colistin is also special with respect to AST, please indicate which guidelines were followed and how Colistin testing was performed
|
||
|
Response 11: Antimicrobial susceptibility testing was conducted using the MicroScan™ NEG MIC 57 panel (Catalog No. C58012; Beckman Coulter®, Brea, California, United States). Following the EUCAST criteria. |
||

Reviewer 2 Report
Comments and Suggestions for Authors
This manuscript presents the first detailed genomic characterization of a clinical isolate of Pseudomonas kurunegalensis, initially misidentified as P. monteilii, from a post-renal transplant patient. Using whole-genome sequencing and a robust comparative genomic approach (ANI, SNP phylogeny), the authors confirm its identity as P. kurunegalensis, highlight its multidrug-resistant phenotype, including the presence of the VIM-2 carbapenemase gene and discuss its clinical and epidemiological relevance. The work contributes significantly to the growing understanding of emerging Pseudomonasspp. in clinical settings, emphasizing the need for genome-based diagnostics. I have suggestions for improving this work as below:
1. The abstract omits specific diagnostic thresholds (e.g., ANI %), lacks clarity on methodology steps, and does not mention that the isolate was from asymptomatic bacteriuria.
2. The research gap and significance are vaguely defined in introduction.
3. Methodological details are extensive but lack organization, especially regarding species identification tools (e.g., BLAST, TYGS, GAMBIT).
4. Resistance mechanism analysis is partially described; membrane-related resistance mechanisms are excluded without justification.
5. No table or visual summary of genome metrics, tools used, or software versions.
6. Missing figures (e.g., Figures 1 and 2 are mentioned but not shown here); weak visualization and summary of phylogenetic/ANI outcomes.
7. Antibiotic susceptibility data lacks comparative control or breakpoint table (e.g., EUCAST/CLSI reference).
8. No mention of plasmid analysis despite VIM-2 presence.
9. The clinical course is described briefly but lacks diagnostic markers (e.g., urinalysis) and host immunological parameters.
10. The discussion repeats known facts (e.g., resistance spread of VIM-2) without deeply contextualizing this strain's novelty or its genomic differences vs. other P. kurunegalensis strains.
11. The discussion blends narrative with results (e.g., ANI %), weakening structure.
12. The manuscript does not address potential limitations of sequencing quality (69 contigs, low N50).
13. Ethical approval is clearly stated, but data availability and genome submission (e.g., GenBank accession number) are not mentioned.
Author Response
|
Response to Reviewer 2 Comments
|
||
|
1. Summary |
|
|
|
Thank you very much for taking the time to review this manuscript. Please find the detailed responses below and the corresponding revisions/corrections highlighted/in track changes in the re-submitted files.
|
||
|
2. Point-by-point response to Comments and Suggestions for Authors |
||
|
Comments 1: The abstract omits specific diagnostic thresholds (e.g., ANI %), lacks clarity on methodology steps, and does not mention that the isolate was from asymptomatic bacteriuria.
|
||
|
Response 1: Thank you for pointing this out. We agree with this comment. In the abstract we have added that the patient had no symptoms of urinary tract infection, the ANI threshold and the steps of the methodology |
||
|
Comments 2: The research gap and significance are vaguely defined in introduction. |
||
|
Response 2: Thank you for pointing this out. We have added the following paragraph in the introduction: Despite the increasing recognition of non-aeruginosa Pseudomonas species in clinical settings, the role and clinical impact of P. kurunegalensis remain poorly understood. To date, there are one report of this species being isolated from human clinical samples. This represents a gap in our understanding of the epidemiology and resistance dynamics of emerging environmental Pseudomonas species that may enter clinical environments.
Between lines 78 and 83. |
||
|
Comments 3: Methodological details are extensive but lack organization, especially regarding species identification tools (e.g., BLAST, TYGS, GAMBIT). |
||
|
Response 3: Perfect, We have added subsections within the methodology. |
||
|
Comments 4: Resistance mechanism analysis is partially described; membrane-related resistance mechanisms are excluded without justification. |
||
|
Response 4: Thank you for pointing this out. Genes of the membrane-related resistance mechanism have very low identity percentages, therefore we cannot ensure the presence of these. Line 202. |
||
|
Comments 5: No table or visual summary of genome metrics, tools used, or software versions. |
||
|
Response 5: Thank you for pointing this out. The genome metrics, tools and versions are written in the methodology section. |
||
|
Comments 6: Missing figures (e.g., Figures 1 and 2 are mentioned but not shown here); weak visualization and summary of phylogenetic/ANI outcomes. Missing figures (e.g., Figures 1 and 2 are mentioned but not shown here); weak visualization and summary of phylogenetic/ANI outcomes. |
||
|
Response 6: Figures are available in the manuscript. I don't know why you don't see them, if you have questions ask the editor. |
||
|
Comments 7: Antibiotic susceptibility data lacks comparative control or breakpoint table (e.g., EUCAST/CLSI reference). |
||
|
Response 7: Thank you for pointing this out. You are absolutely right.We have added a table with antibiotic susceptibility and EUCAST criteria. |
||
|
Comments 8: No mention of plasmid analysis despite VIM-2 presence. |
||
|
Response 8: Thank you for pointing this out. Mob-recon was used to extract the plasmid sequences, and this isolate lacks them, therefore VIM-2 is chromosomal. We have added this tool to the methodology, and the results show that no plasmids were detected. |
||
|
Comments 9: The clinical course is described briefly but lacks diagnostic markers (e.g., urinalysis) and host immunological parameters. |
||
|
Response 9: Thank you for your observation. That information was not included because the manuscript is structured as an original research article rather than a case report. We have discussed this as a team and agreed that the work does not constitute a case report, as its primary focus is on genomic sequencing and analysis.
|
||
|
Comments 10: The discussion repeats known facts (e.g., resistance spread of VIM-2) without deeply contextualizing this strain's novelty or its genomic differences vs. other P. kurunegalensis strains |
||
|
Response 10: This information is found between lines 291 and 298. |
||
|
Comments 11: The discussion blends narrative with results (e.g., ANI %), weakening structure. |
||
|
Response 11: Thank you for your observation.The discussion has been restructured and more information has been added. |
||
|
Comments 12: The manuscript does not address potential limitations of sequencing quality (69 contigs, low N50). |
||
|
Response 12: Thank you for your observation. This information is found between lines 338 and 342. |
||
|
Comments 13: Ethical approval is clearly stated, but data availability and genome submission (e.g., GenBank accession number) are not mentioned. |
||
|
Response 13: Thank you for your observation . We have add the Data Availability Statement: The genome were submitted to GenBank on 16 June 2025 under BioSample ID SAMN48908813. |
||
|
3. Response to Comments on the Quality of English Language |
||
|
The English of the manuscript has been revised |
||

Reviewer 3 Report
Comments and Suggestions for Authors
This report describes the microbiological characterization of a unique member of the Pseudomonas genus. The authors performed good microbiological methods to characterize the isolate. However, a few issues need to be addressed to improve the manuscript.
- Abstract: It would e helpful to add whether or not the patient had symptoms suggestive of a urinary tract infection. This helps distinguishes whether the organism was pathogenic or just a contaminant. Also, was it recovered from a catheterized urine or from a midstream sample?
- Materials and methods: Please indicate the temperature of the incubation.
- Please add the city and country of all the companies listed in the methods sections.
- Results: Did the patient have any UTI symptoms, whether upper (flank pain, fever, nausea,...) or lower (burning micturition, suprapubic pain, perineal pain, ...).
- Was the patient hospitalized when the isolate was recovered? Please clarify.
- It would be better to list the susceptibility and MIC results in a table. Also, please cite the EUCAST Clinical Breakpoints version you used to determine susceptibility breakpoints.
- How come the isolate was susceptible to levofloxacin but the MIC was more than 1 mg/L?
- How did you determine the susceptibility to colistin? Did you use broth microdilution since automated methods are not approved for the susceptibility testing to colistin?
- Discussion: Please elaborate on what could be the potential source of this organism in your patient?
Author Response
|
Response to Reviewer 3 Comments
|
||
|
1. Summary |
|
|
|
Thank you very much for taking the time to review this manuscript. Please find the detailed responses below and the corresponding revisions/corrections highlighted/in track changes in the re-submitted files.
|
||
|
2. Point-by-point response to Comments and Suggestions for Authors |
||
|
Comments 1: Abstract: It would e helpful to add whether or not the patient had symptoms suggestive of a urinary tract infection. This helps distinguishes whether the organism was pathogenic or just a contaminant. Also, was it recovered from a catheterized urine or from a midstream sample?
|
||
|
Response 1: Thank you for pointing this out. We agree with this comment.
We have added “recovered from a catheterized urine sample of a post-renal transplant patient without symptoms of urinary tract infection” It is located on lines 18 and 19.
|
||
|
Comments 2: Materials and methods: Please indicate the temperature of the incubation.
|
||
|
Response 2: Thank you for pointing this out. We have, accordingly, add this. “and 24 hours incubation at 35ºC in aerobic atmosphere.” it can be found at line 94 |
||
|
Comments 3: Please add the city and country of all the companies listed in the methods sections.
|
||
|
Response 3: Thank you for pointing this out. We have added the cities and countries of each of the commercial houses. These changes are located between lines 96 and 113. |
||
|
Comments 4: Results: Did the patient have any UTI symptoms, whether upper (flank pain, fever, nausea,...) or lower (burning micturition, suprapubic pain, perineal pain, ...). |
||
|
Response 4: Thank you for pointing this out. The patient was asymptomatic, so we've added the following sentence: "At the time of sampling, the patient was asymptomatic" on line 153 |
||
|
Comments 5: Was the patient hospitalized when the isolate was recovered? Please clarify.
|
||
|
Response 5: We add that the patient was not hospitalized, this can be found on line 153-154 |
||
|
Comments 6: It would be better to list the susceptibility and MIC results in a table. Also, please cite the EUCAST Clinical Breakpoints version you used to determine susceptibility breakpoints.
|
||
|
Response 6: The table with the susceptibility was created, it is located on line 159 and the EUCAST Clinical Breakpoints version in line 102 |
||
|
Comments 7: How come the isolate was susceptible to levofloxacin but the MIC was more than 1 mg/L?
|
||
|
Response 7: It was an error, this modification is already reflected in the table. |
||
|
Comments 8: How did you determine the susceptibility to colistin? Did you use broth microdilution since automated methods are not approved for the susceptibility testing to colistin?
|
||
|
Response 8: Antimicrobial susceptibility testing was conducted using the MicroScan™ NEG MIC 57 panel (Catalog No. C58012; Beckman Coulter®, Brea, California, United States). Following the EUCAST criteria. |
||
|
Comments 9: Discussion: Please elaborate on what could be the potential source of this organism in your patient?
|
||
|
Response 9: This can be found between lines 257 and 263. |
||

Round 2
Reviewer 2 Report
Comments and Suggestions for Authors
I have no any comments for this revised version.
Reviewer 3 Report
Comments and Suggestions for Authors
The authors did a good job addressing the comments. One monro comment during the proofreading step: References 10 and 11 need to be edited for version and access date. Also, replace the French "Disponoble en" to "Available from" or "Available at"